# A Review of Unmanned Aerial Vehicle Based Antenna and Propagation Measurements

**DOI:** 10.3390/s24227395

**Published:** 2024-11-20

**Authors:** Venkat R. Kandregula, Zaharias D. Zaharis, Qasim Z. Ahmed, Faheem A. Khan, Tian Hong Loh, Jason Schreiber, Alexandre Jean René Serres, Pavlos I. Lazaridis

**Affiliations:** 1School of Computing and Engineering, University of Huddersfield, Huddersfield HD1 3DH, UK; q.ahmed@hud.ac.uk (Q.Z.A.); f.khan@hud.ac.uk (F.A.K.); p.lazaridis@hud.ac.uk (P.I.L.); 2School of Electrical and Computer Engineering, Aristotle University of Thessaloniki, 54124 Thessaloniki, Greece; zaharis@auth.gr; 3Electromagnetic & Electrochemical Technologies Department, National Physical Laboratory, Teddington TW11 0LW, UK; tian.loh@npl.co.uk; 4Sixarms, West Burleigh, Burleigh Heads 4219, Australia; jason@sixarms.com; 5Center of Electrical Engineering and Informatics, Universidade Federal de Campina Grande, Campina Grande 58708-110, Brazil; alexandreserres@dee.ufcg.edu.br

**Keywords:** absorbers, broadcasting systems, knife edge diffraction (KED), parabolic reflector, path loss, propagation measurements, square kilometer array (SKA), UAV-based measurements, unmanned aerial vehicles (UAVs)

## Abstract

This paper presents a comprehensive survey of state-of-the-art UAV–based antennas and propagation measurements. Unmanned aerial vehicles (UAVs) have emerged as powerful tools for in situ electromagnetic field assessments due to their flexibility, cost-effectiveness, and ability to operate in challenging environments. This paper highlights various UAV applications, from testing large–scale antenna arrays, such as those used in the square kilometer array (SKA), to evaluating channel models for 5G/6G networks. Additionally, the review discusses technical challenges, such as positioning accuracy and antenna alignment, and it provides insights into the latest advancements in portable measurement systems and antenna designs tailored for UAV use. During the UAV–based antenna measurements, key contributors to the relatively small inaccuracies of around 0.5 to 1 dB are identified. In addition to factors such as GPS positioning errors and UAV vibrations, ground reflections can significantly contribute to inaccuracies, leading to variations in the measured radiation patterns of the antenna. By minimizing ground reflections during UAV–based antenna measurements, errors in key measured antenna parameters, such as HPBW, realized gain, and the front-to-back ratio, can be effectively mitigated. To understand the source of propagation losses in a UAV to ground link, simulations were conducted in CST. These simulations identified scattering effects caused by surrounding buildings. Additionally, by simulating a UAV with a horn antenna, potential sources of electromagnetic coupling between the antenna and the UAV body were detected. The survey concludes by identifying key areas for future research and emphasizing the potential of UAVs to revolutionize antenna and propagation measurement practices to avoid the inaccuracies of the antenna parameters measured by the UAV.

## 1. Introduction

Unmanned aerial vehicles (UAVs) or remotely piloted aircraft systems, simply known as drones, are in high demand for in situ measurements because of their mobility, low cost, hovering capability, and low maintenance expenses. Advances in technology, such as software–defined radios (SDRs), have facilitated the utilization of UAVs for antenna and propagation measurement [1,2]. UAVs are an effective antenna measurement solution for projects such as the square kilometer array (SKA) [3], broadcasting systems [4], large biconical antennas [5], such as those of the French aerospace research center Office National d’Etudes et de Recherches Aérospatiales (ONERA) [6], and parabolic reflector antennas [7], because of their in situ electromagnetic field measurement capabilities. UAV measurements can also help to identify multipath–related propagation losses. In radar applications that use antenna arrays, UAV measurements are useful for the final calibration after the antennas are enclosed in radomes and deployed. This is particularly crucial in cases in which the RF system is large. Climatic conditions, such as rain, snow, or harsh weather, degrade the antenna radiation performance in terms of side–lobe level (SLL) and ripples in the beam peak. The degradation of these antenna parameters can eventually lead to a deterioration of the antenna gain and cross-polarization levels [8]. Conventional methods may not be feasible in such situations, making UAV–based measurements a viable alternative.

In practice, antenna measurements with UAVs can be performed either in the Fresnel region (also known as the radiative near field) or in the Fraunhofer region (also known as the far field). For example, in shortwave communications, an antenna operating at 20–30 MHz can achieve very long communication distances owing to ionospheric propagation. Nevertheless, in this frequency range, the antenna size is large. To avoid large separation distances between the antenna under test (AUT) and the UAV, near–field measurements are an appropriate and practical alternative, whereby the near–field measurements are mathematically transformed into far–field radiation patterns. In this paper, the utilization of UAVs to perform near–field measurements is presented and involves the UAV following a specific spatial path [9,10,11] to collect measurements of the AUT using an SDR.

Measuring antenna performance using the traditional outdoor method [12] involves measuring the antenna in the far–field region. Accordingly, these measurements require several pieces of expensive and heavy equipment. The AUT is in the receiving mode and is connected to a mixer and microwave receiver [13], which can retrieve the amplitude and phase information of the AUT. Supported by model towers, the AUT is placed on a multi-axis rotary positioner, such as an azimuth–over–elevation-over-azimuth positioner [14]. These positioners can rotate the AUT by 360° in the azimuth plane and provide limited motion in the elevation plane.

Unlike traditional outdoor measurements, UAV–based measurements do not rely on large or heavy equipment. For example, instead of a heavy spectrum analyzer, an SDR, which is compact and lightweight, can be used. In addition, these SDRs can be controlled from ground level. Taking advantage of UAVs’ capability to hover and perform circular trajectories would make them free from the positioners and controllers used in the traditional measurement methods. In addition to the UAV–based antenna measurements, UAVs can also be used for propagation measurements. UAVs equipped with SDRs can be used to perform propagation measurements, which involve estimating signal strength as a function of frequency and time. These measurements are then post–processed to extract the channel coefficients, such as path loss and angle of arrival, which are used to model the channel. The channel model can then be used to design an air–to–ground (ATG) communication system [15] and evaluate its performance under different operating conditions. UAVs are used to evaluate the performance of existing ATG communication systems and to identify potential sources of interference or signal degradation [16]. Propagation measurements can be conducted by mounting a transmitting antenna and lightweight transmitter with a built–in battery on a UAV and a standard calibrated antenna acting as a receiver connected to a spectrum analyzer at the ground level. A critical issue is airframe shadowing, which obstructs the line–of–sight (LOS) path between the transmitter and receiver caused by the UAV body. In this regard, it is important to verify that the transmitting antenna mounted on the UAV is in direct LOS with the receiver before performing any propagation measurements [17].

This article describes how UAVs equipped with an antenna and an SDR configured as a receiver can be used to measure antenna and propagation characteristics in various practical cases. In contrast, previous studies have emphasized only one specific case each time. The main contributions of this study are in the following topics:A comprehensive review of the latest advancements in UAV–based antenna and propagation measurements encompassing various techniques and applications.Implementation of both near–field and far–field measurement techniques, emphasizing the practical advantages, including a discussion on the use of SDRs in UAV–based measurements.Exploration of a wide range of applications, from the SKA and large–scale biconical antennas to 5G/6G network evaluations.Illustration of practical implementations and effectiveness of UAV–based measurement systems, presented using real–world test cases, such as measurements of parabolic reflector systems and large–scale propagation channel effects.

The remainder of this paper is organized as shown in Figure 1. Section 2 focuses on UAV–based measurements. In this section, we provide design recommendations for UAV–based measurements. By following the design recommendations, we describe how UAV–based measurements can be performed in the far– and near–field regions. For UAV–based antenna measurements in the far–field region, we describe the procedure for measuring base station antennas (BASTAs) and digital television (DTV) stations deployed in the field. To measure antennas operating at lower frequencies, a larger distance between the AUT and UAV is required, owing to the large dimensions of the AUT. To overcome these large separation distances, antenna measurements can be performed in the near–field region.

We explain UAV–based near–field measurements for structurally large antennas, such as low–frequency aperture arrays (LFAAs) and biconical antennas. UAV–based propagation measurements are presented in Section 3. In this section, a detailed explanation of the simulation results for large–scale and small–scale propagation measurements is presented. Based on the UAV–based antenna and propagation measurements, we explain the design considerations for selecting the UAV probe, and the conclusions are provided in Section 4.

## 2. UAV–Based Measurements

Characteristics such as the matching impedance, polarization, radiation efficiency, directivity, gain, and radiation patterns are used to ascertain the antenna performance. For radar systems with stringent specifications, such as polarimetric weather radar, calibration of the systems after deployment in the field is crucial. Figure 2 describes the methodology adopted for UAV–based measurements, in which the UAV is equipped with an antenna at a height *h_uav_* that serves as a signal source, while the AUT is at a height *h_aut_* from the ground plane. An antenna mounted on a UAV can act as either a transmitting (TX) or a receiving (RX) antenna. As shown in Figure 2, the AUT and the TX antenna mounted on the UAV were separated by distance *R*. Here, *α* denotes the half–power beam width (HPBW) of the TX antenna mounted on the UAV, and *α_h_* denotes the plane angle subtended at the antenna mounted on the UAV by the AUT height [18].

In the process of measuring the radiation pattern of an AUT using a UAV, several factors, such as the phase curvature of the incident wave, ground reflections, and amplitude taper of the source antenna, affect the measurement accuracy. To avoid measurement inaccuracies, caution should be exercised regarding the variations in the phase and amplitude of the incident field and the interference from ground reflections. For UAV–based measurements, as a rule of thumb, the TX antenna is selected such that it has a wider HPBW than the AUT to prevent measurement errors. Similarly, the phase curvature of the incident field on the AUT affects the accuracy of the measured SLL. To overcome these errors, the phase deviation over the planar test aperture is maintained below 22.5°. To achieve this phase deviation, it is necessary to have a separation distance greater than 2*D*^2^*/λ*, where *D* is the maximum dimension of the antenna and *λ* is the operating wavelength [19]. The other factors that affect the accuracy of the measurements are ground reflections.

Ground reflections can often cause signals to be added constructively/destructively, resulting in apparent gain values that are higher or lower than expected. Also, maintaining a sufficient distance between the TX and RX antennas prevents the distortion of the patterns caused by ground reflections. Different approaches have been used to reduce ground reflections, including diffraction screening and absorbers between the TX and RX antennas. The use of a TX antenna with a low SLL can also prevent ground reflections [20]. At the same time, aligning the beam peak of the TX antenna with the AUT is essential. This can be achieved in a couple of iterations, which involve measuring the power received at the AUT when a UAV is carrying the TX antenna that flies at different altitudes. At a particular height, the UAV can further vary its altitude slightly to detect the position corresponding to the maximum received power of the AUT. At this height, the beam peak from the TX is appropriately aligned with the AUT. With the TX aligned with the AUT and by maintaining a low SLL, ground reflections can be avoided to a certain extent.

### 2.1. UAV–Based Far–Field Measurements

To perform far–field measurements, it is essential to maintain a minimum distance of 2*D*^2^*/λ*, as discussed previously. A UAV equipped with high–precision controller boards and RF measurement equipment, such as an SDR [21] with directional antennas, can be used for far–field measurements. For instance, in the inspection of reflector systems [7] and structurally large antennas, UAV–based measurements are cost–effective and reliable. The measurement techniques proposed here have several advantages over conventional methods, such as the use of helicopters equipped with RF payloads [22], in terms of cost and maneuverability. A compact and lightweight design allows UAVs to easily reach any location for measurements. They can also hover at a specific location, which enhances their ability to conduct RF measurements, with improved results. Various techniques, such as fast Fourier transform (FFT) [23], angular deconvolution [24], spatial mode filtering [25], frequency impulse response, and Hilbert transform, can be used to filter noise when measurements are conducted outdoors in a noisy environment. Considering that the above methods are not generic and cannot be applied to all environmental conditions, [26] proposed a filtering technique referred to as locally weighted regression and dispersion smoothing, which can be used to filter out high–frequency noise.

To validate this methodology, measurements were conducted in an anechoic chamber as well as an outdoor environment using the proposed filtering technique on a Yagi–Uda antenna operating at 2.4 GHz and a horn antenna at 5.3 GHz. An analysis of the HPBW measured in an anechoic chamber and an outdoor environment showed that they were in good agreement, with a difference of only 1°. In the far–field measurements, the far–field patterns on a sphere of a constant radius were estimated. The elevation and azimuth angles, denoted by *θ* and *ψ*, were the variables used to identify the location on the sphere. The phase information of the AUT was obtained using a vector network analyzer (VNA), and the two-dimensional amplitude information may be calculated using the total electric field:(1)$$\left|E\right|=\sqrt{{\left(E_{\theta }\right)}^{2}+{\left(E_{\psi }\right)}^{2}},$$
where *E_θ_, E_ψ_* are the electric field in the elevation and azimuth planes, respectively.

In UAV–based far–field measurements, the AUT is placed on a tripod and the UAV follows a vertical and horizontal path around the AUT. The UAV carries an RX antenna while moving and collecting data points. The AUT is stationary in this scenario, unlike the conventional outdoor ranges, which can be either elevated or elevated slant ranges, and requires the AUT to rotate by means of positioning commands operated by a computer.An example of a conventional elevated slant range is shown in Figure 3, in which a TX antenna, such as a quad–ridged horn, is mounted on a tall structure, and an AUT, such as an offset–fed parabolic reflector system, is mounted on an azimuth positioner. To eliminate errors caused by ground reflection, absorbers are placed between the source antenna and the AUT. Generally, a TX antenna is designed to have a low SLL, and the height at which the source antenna is mounted should be selected such that the main beam illuminates the AUT. Traditional measurements are performed using a stationary source antenna, and the radiation pattern can be obtained in both the elevation and azimuth cuts by rotating the AUT placed on the positioner. However, in UAV–based measurements, the UAV carrying the antenna follows a trajectory and the AUT is stationary. The advantage of this setup is that it does not require heavy and expensive positioners to obtain AUT radiation patterns.

As shown in Figure 4, a UAV with a quad-ridged dual–polarized horn operating at 6–24 GHz was used as a far–field transmitting antenna [27] to measure the microwave vision group (MVG) SR40 parabolic reflector system. During the outdoor measurements, the UAV and AUT were maintained at 350 and 750 m, respectively. The TX antenna mounted on the UAV was supported by a gimbal, which was used to detect radiation patterns in the elevation plane, and the rotation of the UAV around the parabolic antenna was performed to measure the radiation patterns in the azimuth plane. All the measurements were performed at 14.5 GHz. During UAV measurements, errors may occur because of external winds, reflections from surfaces, and misalignment between the probe and the AUT caused by UAV propeller vibrations. To minimize these errors, various methods are employed, such as measurement of the AUT under additional conditions, such as rotating the device at 180°, conducting measurements with different separation distances, and taking multiple measurements at a time in one cut and averaging them.

Figure 5 illustrates how UAVs can measure the performance of antennas mounted on ships. In [28], a UAV carrying a vertically polarized ground plane monopole antenna and three vertical radials was used as an RX antenna to measure an X–band vertically polarized reflector antenna mounted on a ship that resonates at 9.5 GHz. Throughout the UAV flight, the antenna mounted on the UAV is directed towards the AUT; however, the variance in the pitch and roll axes while the UAV is in motion creates a polarization mismatch. Based on the experiments conducted in [28], a loss of 0.2 dB is observed in the measured radiation pattern due to polarization mismatch. Traditionally, to measure the performance of an antenna installed on a ship, an RX antenna connected to a spectrum analyzer is required to collect the signals. The RX antenna is placed at the ground level on the shore. The reflector antenna, which is the AUT installed on the ship, continuously transmits signals. To measure the radiation pattern of the AUT in the azimuth plane, the ship carrying the reflector antenna should follow a circular trajectory in the sea, while the receiver collecting the signals is static at ground level. Conversely, for UAV–based measurements, an SDR mounted on the UAV is used to receive the signals. A reflector antenna installed on the ship transmits signals. Here, the ship carrying the TX is static, and the UAV carrying the RX antenna follows a circular trajectory with a constant radius around the ship situated at the center of the circle. When measuring an antenna installed on a ship, it is essential to meet specific criteria to ensure that there are no losses due to polarization mismatches [29] or multipath reflections from seawater [30]. Considering that the AUT has a maximum diameter of 1 m and operates at 9.5 GHz, it is imperative to maintain a far–field distance (Fraunhofer distance) greater than 60 m, and no obstacles should block the first Fresnel zone between the ship and the UAV.

#### 2.1.1. Aerial Measurement of Base Station Antennas

To establish communication, a mobile BASTA [31] is essential and serves as a communication hub for wireless devices. Owing to the exponential increase in the number of devices connected to wireless networks, BASTAs are being deployed at an unprecedented rate to provide connectivity to users. Certain errors may occur when BASTAs are deployed in the field. These errors include undesired antenna twists, antenna tilts, errors in antenna alignments, and the effects of adjacent objects and towers on radiation patterns. In such cases, performing an in–situ measurement allows one to identify faults and repair the system, thereby improving its performance. In traditional airborne measurements, a helicopter is used to measure the radiation pattern. However, these methods are expensive and require heavy equipment. Advancements in UAVs and the miniaturization of RF components, such as portable spectrum analyzers, have enabled the measurement of mobile BASTA systems using UAVs [32].

UAV–based measurements for BASTAs involve measuring vertical and horizontal radiation patterns. The vertical radiation pattern is determined using the procedure shown in Figure 6. To ensure optimal reception from the AUT, the UAV should be positioned at an appropriate height to maintain LOS with the BASTA. Once the optimal vertical location is determined, the UAV follows a vertical path and reconstructs the radiation pattern. Typically, a BASTA consists of several antenna elements, such as dipoles, arranged in an array. Each of these elements has its own radiation pattern. The resultant radiation pattern is formed by combining all the elements in an array. Therefore, the UAV must follow a vertical path to reconstruct its vertical radiation pattern. The electrical down–tilt and null fill can be determined by obtaining the vertical radiation pattern of the BASTA. Electrical down–tilt [33] and null fills are significant parameters that affect the base station coverage area. Ground users experience maximum signal strength when the main lobe is directed towards their area, whereas they do not receive any signal when a null fill is directed towards them. In [31], a UAV programmed with a mask R–CNN was used to automatically determine the base station orientation. R–CNN is an object detection algorithm used to detect specific regions in an image. The proposed method in [31] involves creating a database named UAV–antenna, which consists of 19,715 communication BASTA images. This is achieved by the capturing of BASTA images by UAVs. Secondly, mask R–CNN applies a selective search scheme to identify the pixel coordinates of the BASTA. These pixel coordinates are used to measure the BASTA’s tilt angle. Based on the proposed method, after completing the measurements it was found that the actual tilt angle of the antenna system deviated by 1°–2° from the intended tilt angle.

Figure 7 illustrates the procedure followed to obtain the horizontal radiation pattern of a BASTA deployed in the field, which was measured using a UAV with an RX antenna. Based on the optimum height, at which the RX antenna mounted on the UAV receives the maximum power from the AUT, which is determined during the vertical pattern measurements, the UAV follows a trajectory in a circular path around the AUT. Obtaining a horizontal radiation pattern enables one to determine azimuth HPBW, sector power ratio (SPR), and front-to-back ratio (FBR). SPR is the ratio of power outside the desired sector to power inside the desired sector. This helps to improve the antenna design, which requires the SPR to be as low as possible to achieve lower co–channel interference and better call quality. Ideally, a BASTA should have an SPR less of than 3% and an FBR greater than 25 dB [34].

#### 2.1.2. Aerial Measurement of Broadcasting Antennas

In [35], a SixArms custom-built hexacopter with a log–periodic antenna was used to perform broadcasting antenna measurements in the far-field region. It was used to measure the effective radiated power (ERP) and horizontal radiation pattern (HRP), as well as the vertical radiation pattern (VRP). As shown in Figure 8, high–power broadcast antennas can experience certain deviations in their performance, leading to a degradation in the overall coverage area. The feeding mechanism of a broadcast antenna [36] plays a significant role, and these systems are vulnerable to changes. Upon performing the in–situ measurements and comparing the measured VRP with the design specifications, it was observed that there was a 1° deviation in the tilt in the test case of [32]. Similarly, a change of 0.5° in the electrical tilt was observed owing to the change in the mechanical lean of the broadcast antennas. By measuring the HRP, other common errors, such as incorrect panel orientation and inverted panels, could be identified and eliminated, thereby enhancing the overall performance of the broadcast antenna [37]. By comparing the HRP of the broadcasting antenna measured with the UAV with the design specifications, it became apparent that a 10° deviation arose in this test case from an incorrect panel orientation after the panel was installed on the tower. Similarly, measuring the HRP of the broadcast antenna, when there was a taller tower at just 650 ft, showed a 3 dB notch in the plots, which implies that the adjacent tower impacts the measured system.

Ideally, when amplitude–only measurements are performed, UAVs carrying transmitters/receivers are in the far–field region (the Fraunhofer region) to measure the radiation pattern [20]. To validate the theoretical concept of far–field regions, the SixArms Airborne Radio Measurement Systems (ARMS-RFX) UAV was used to measure a DTV station at 720 m and 2025 m from the AUT. DTV broadcasting antennas are composed of antenna arrays formed by similar elements. The total height of a DTV broadcasting antenna with all the elements in an array is typically 20 m, with a maximum antenna dimension of 20 m and a frequency of operation at 515 MHz (UHF channel 21 in the USA). Far–field measurements with UAVs can be performed by maintaining at least 1450 m from the AUT. As shown in Figure 9, ARMS–RFX UAVs equipped with an ARMS receiver comprising a real–time spectrum analyzer and an embedded PC [38] were used to measure the DTV transmitter station at 720 m and 2025 m from the AUT. Log–periodic antennas (LPDA) mounted on top of the UAV received signals from the AUT. The radiation pattern of the AUT was calculated instantly for every 0.1°, as the UAV took a vertical path. Using a telemetry link, the measured radiation patterns were transmitted to the ground user for quick verification.

The elevation patterns taken at 720 m, which should have been 1450 m according to the theoretical far–field distance calculations, and at a far–field distance of 2025 m from the AUT, are depicted in Figure 10. In Figure 10a, the red dashed lines represent the elevation pattern of the AUT according to the manufacturer’s data sheet, and the solid black line represents the measurements performed with the UAV at 720 m. Similarly, Figure 10b represents the measurements performed at 2025 m from the AUT. From the elevation patterns measured at 720 and 2025 m, it was observed that the measurements do not always have to be in the far–field region. UAV–based measurements can be performed closer to the AUT and are still valid. From the two cases depicted in Figure 10, it is evident that when measurements are performed at 720 m from the AUT, the null fill and null depths vary slightly compared to measurements performed at 2025 m from the AUT. In the following sections, we discuss UAV–based antenna measurements in near–field regions. This technique enables the assessment of antennas that are significantly large. Additionally, conducting measurements in the near–field offers the advantage of reducing the flight times required for UAV operations. In Table 1, we present an overview of the reflector and broadcasting antennas measured in far–field regions using UAVs.

### 2.2. UAV–Based Near–Field Measurements

To meet the far-field criteria, low-frequency antennas require a considerable distance between the AUT and UAV carrying the antenna. Such long distances may result in excessive free-space path loss, which can reduce overall system accuracy. In such cases, near-field measurements can be useful for EM wave measurements in the radiative near-field region. It is easy and quick to conduct near-field measurements with a UAV and does not require heavy and complicated equipment. In [39], a biconical antenna operating at 110 MHz was measured using a UAV under indoor conditions. Time domain gating techniques were applied to avoid ground reflections and UAV motion. Similarly, in [40], navigation systems were measured in the near–field region. The measurement results from [39,40] prove that UAV–based near–field measurements can be performed accurately with low–cost equipment.

As shown in Figure 11, UAV–based measurements were performed at radiating near–field regions beginning at 2.5 m for a grid reflector with a maximum diameter of 1 m that operates at 4.65 GHz [41]. Because the accuracy of UAV–based near–field measurements depends on UAV coordinates, [42] used dual–band real–time kinematics (RTK), which made it possible to obtain UAV coordinates within 10 arcseconds. The UAV was equipped with two monopoles operating in the 4–7 GHz frequency band that were separated from each other by a space of 80 cm. This setup measured a reflector system operating at 4.65 GHz fed by a circularly polarized helix antenna. The monopoles were mounted on a UAV using three–dimensional (3D) printed components, which are highly resistant to mechanical vibrations and transparent to EM waves. Near–field measurements were performed by obtaining the equivalent current distribution over the surface of the AUT. A phase–less retrieval technique was utilized to plot the radiation pattern based on the integral equation method [43], in which the simulated AUT was modeled with equivalent electric and magnetic fields on a closed surface. After obtaining the radiation patterns for the reflector system deployed in the field, measurements were conducted in anechoic chambers. Comparisons between the results obtained from the UAV–based near–field measurements and the measurements in anechoic chambers indicate that when the reflector system is deployed in the field, there is an offset in the beam position and the main beam is widened owing to misalignment errors and ground reflections.

In Figure 12, a UAV carrying a monopole is used to characterize an antenna operating at 3–30 MHz. The AUT is a Nostradamus ONERA system [44] consisting of a set of 288 biconical antennas arranged along a branch separated by 120°. Biconicals are omnidirectional, and each has a height of 7 m and a width of 6 m. To validate the accuracy and functionality of UAV–based near–field measurements for characterizing the high–frequency (HF) antenna by ONERA, ref. [6] used a Dà-Jiāng Innovations (DJI) Matrice 600 Pro carrying an antenna to measure a monopole that was 6 m high when placed on a ground plane. The measured data were compared with simulated data, and good agreement was achieved, thus proving that UAV–based near–field measurements are a cost–effective solution for characterizing HF antenna systems.

When selecting the appropriate material for mounting the antenna on the UAV, it is important to ensure that the UAV body does not degrade the performance of the antenna. In [45], a Mikrokopter equipped with a signal generator and a short monopole was employed to measure a 6 m parabolic dish fed by a dual–polarized LPDA operating in the frequency range of 300 MHz to 3 GHz. In this UAV model, the ground plate was made of aluminum, and to minimize the impact of the UAV body on the antenna, a mesh structure was placed between the frame of the UAV and the antenna. Similarly, the UAV propellers can generate harmonics that are influenced by the propellers’ rotations per minute (RPM) and their dimensions. The Doppler spectrum and harmonics [46] generated by the propellers can be studied using the double–edge diffraction model [47]. It was observed that when an antenna mounted on a UAV transmits signals while the propellers rotate, the signal received by the AUT exhibits a Doppler effect. The Doppler effect, harmonics, and scattering experienced by EM waves from an antenna lead to a drop in the power received by the AUT, which affects the radiation pattern of the AUT. The effects of propellers can be avoided by placing the antenna on the UAV at a location which is far from the propellers. Other approaches, such as using fiber glass material instead of carbon fiber for the propellers, are validated in [48,49]. All the measurements of the radiation patterns in this case were performed at 328.5 MHz.


**Aerial Measurements of Low–Frequency Antennas**


The SKA is an array of telescopes that operates based on the principle of aperture synthesis and is designed for excellent spatial and angular resolution. A square kilometer array log–periodic antenna (SKALA) is a very large structure, and measuring such a large structure in the far–field region requires a large measurement distance. Performing UAV–based far–field measurements for these antennas is not economical owing to the battery limitations of UAVs. Sometimes, the UAV altitude needs to be more than 120 m, which is not possible owing to UAV flying guidelines. In such cases, the antennas are measured in near–field regions. One such case is the SKALA; it consists of 256 LPDAs with a diameter of 38 m. In the SKALA, each LPDA comprises a bowtie dipole for impedance matching. In [50], a pre–aperture array verification system for SKA was measured with UAVs in the near field. The AUT consisted of 16 active elements. All the elements were designed to feature a dual–polarization operation in the frequency range of 50–350 MHz, a minimum directivity of 8 dBi, and an intrinsic cross–polarization ratio exceeding 15 dB. An inter–element spacing of *λ*/2 was maintained to achieve better control over beam steering. However, maintaining an inter–element spacing of *λ*/2 is subject to mutual coupling [3].

The pre–aperture array verification system of SKA, which is the AUT, has an overall size of 9.2 m over a 16 m ground plane mesh. The metallic grid ground plane improved the overall directivity of the system and provided protection from humidity, weather, and terrain conditions. The UAV was equipped with a portable signal generator and a dipole resonating at 175 MHz. The main challenge in these measurements is obtaining accurate phase values. In [50], to address this problem, an additional reference antenna with a known phase [51] was used. This reference antenna was placed 12 *λ* (20 m) from the center of the array, as shown in Figure 13. The UAV was equipped with a dipole and followed a quasi–planar trajectory at an altitude of 24 m. With an average speed of 3 m/s, the UAV took 15 min to complete the trajectory.

When the UAV follows the trajectory, the receiver system connected to the AUT at the ground level acquires voltages corresponding to the horizontal and vertical directions. The time stamps of the global navigation satellite systems (GNSSs) on the UAV and the time stamps of the GNSS at the ground level are synchronized at the receiver connected to the AUT. Finally, with the complex voltages and phase values, the embedded element patterns are reconstructed by performing a near–field to far–field transformation. The measured data from the UAV were compared with the simulated data, and the simulations were performed in the CST studio suite. From these comparisons, it was observed that the UAV–based near–field measurements were accurate. A deviation of 1 dB was observed in the amplitude. Thus, UAVs can be used to measure large structures such as the SKA in the near–field region. Thus, UAV–based measurements for the SKALA can help to identify areas where design improvements are required to improve the efficiency of the entire system. Table 2 presents a summary of large antennas, such as reflector antennas, ground plane antennas, ONERA biconicals, and SKALA LPDAs measured by UAV near–field techniques.

## 3. UAV–Based Propagation Measurements

With advancements in technology and the demand for wireless connectivity, especially for UAVs and other applications, there is a great demand for wireless networks with low latency. These networks require a latency as low as 1 ms, which is a significant improvement over the 40 ms latency of fourth–generation (4G) networks. Transmission needs to be moved to millimeter wave (mmWave) or even terahertz (THz) frequencies to achieve such low latency. EM waves experience higher propagation losses at such high frequencies, owing to diffraction [53] and scattering from rough surfaces. Therefore, understanding the propagation environment through propagation measurements [54] is essential. The propagation of EM waves in an environment can be evaluated by using UAV–based propagation measurements. In addition, UAV–based propagation measurements enable the measurement of key performance indicators (KPIs), such as reference signal received quality (RSRQ) and reference signal received power (RSRP). In [55], a hexacopter carrying a smartphone, sensors, and guided autonomous flight paths was used to measure RSRP and RSRQ. During the transmission, the signal was attenuated by various factors before it reached the receiver. The most common reasons for signal attenuation are path loss, shadowing, and multipaths. Shadowing and multipath components, such as reflection, refraction, diffraction, and scattering, are primarily caused by obstacles. In sub-Section 3.1, we discuss large–scale and small–scale propagation in detail.

### 3.1. UAV–Based Large–Scale Propagation Measurements

UAVs can be used to measure large–scale propagation effects that occur mainly owing to path loss and shadowing. To estimate the path loss, the commonly used models are the free–space path loss (FSPL), two–ray, basic log–distance, and modified log–distance models. For the path loss calculation, the FSPL requires information about the transmitter antenna gain (*G_t_*), receiver antenna gain (*G_r_*), operating wavelength (*λ*), and separation (*d*) between the UAV and receiver. On the other hand, when the UAV is at a lower altitude and ground reflections are present between the UAV and receiver, two–ray models can be utilized to estimate the path loss. In [56], a DJI Mavic 2 Enterprise UAV equipped with a LoRa sleeve dipole operating at 868 MHz was used for propagation measurements between a UAV and a wireless sensor network (WSN) and between a UAV and an unmanned surface vehicle (USV). The results of the UAV–based propagation measurements for a scenario in which the UAV moved vertically up to 30 m, and another scenario in which the UAV moved horizontally away from the receiver (WSN or USV), were compared with path loss estimations from two–ray models. Based on the comparison, the two–ray model underestimated the path loss, resulting in a mean difference of 6.45 dB between the UAV and USV and 15.5 dB between the UAV and WSN. These findings demonstrate the necessity of improving the two–ray model to increase the accuracy of path loss measurements. A higher level of precision is required for mmWave [57,58,59] and for situations in urban areas [60], which are surrounded by multiple buildings and obstacles.

Log–distance path loss models are more general and appropriate for calculating path loss [61]. In evaluating the path loss for a channel between a UAV and a receiver, the log–distance method considers the path loss exponent (*α*). An improvement in the log–distance model is the modified log–distance model. According to the modified log–distance model [62,63], we can determine the path loss by
(2)$$PL\left(dB\right)={PL}_{0}\left(dB\right)+10\alpha {log}_{10}\left(\frac{d}{d_{0}}\right)-10{log}_{10}\left(\frac{∆h}{h_{opt}}\right)+C_{p}+10{log}_{10}\left(1+\frac{∆f}{f_{c}}\right),$$

When estimating the path loss, the modified log–distance model considers an additional parameter known as the height of the UAV from the ground (*h_gnd_*), the minimum height of the UAV (*h_opt_*) that provides the lowest path loss, and a constant loss factor *C_p_*, representing the losses due to the antenna orientation on the UAV and carrier frequency *f_c_*. *PL*_0_ (dB) is the path loss at the reference distance *d_0_*, Δ*f* is the Doppler variation in frequency, and Δ*h* is the difference between *h_gnd_* and *h_opt_*.

To determine the extent to which the various factors discussed above attenuate the signals, we created a scenario with a UAV carrying the transmitting antenna. The propagation measurements [64], as described in Figure 14, consisted of a car and horn antenna that was identical to that mounted on a UAV. The ground–level antenna worked in the receive mode and was mounted on a 2 m mast in front of a 10 m building. The horn antenna was mounted on the UAV hovering 12 m above ground level. Modeling and simulations were performed using commercially available EM software, CST Studio suite 2023 [65] at 5 GHz. Part of the signal was diffracted by the corners of the buildings at a height of 10 m and by metallic components of masts and obstacles, such as cars, before reaching the receiver at ground level.

In [66], a UAV equipped with a dipole and scanner capable of measuring RSRP was utilized to calculate the path loss, with the UAV’s altitude varying from 1.5 m to 120 m above ground level. To understand the influence of LOS and non–line–of–sight (NLOS) conditions on the path loss calculations in [67], the path loss was measured in an area of 500 m × 500 m. As part of this experiment, propagation measurements were performed in an urban area in Greece, which consists of buildings and trees. Measurements were conducted at 2.12 GHz for both LOS and NLOS scenarios, with the UAVs operating at an altitude of 6 to 200 m above ground level. The attenuation caused by reflections and diffractions from buildings and obstacles was also considered for the path loss calculations using the log–distance path loss model. The calculations were based on the uniform theory of diffraction (UTD) and geometric optics (GO). The path loss calculations varied for the LOS and NLOS conditions. The UAV flew at 100 m above the ground; for the LOS condition, *α* was as low as 2.6; however, at the same altitude, for the NLOS condition, *α* was 7.2, indicating that the path loss values differed depending on the test conditions.

The polarization mismatch of the antenna mounted on the UAV also affects the accuracy of the path loss measurements. In [68], a DJI Phantom 4 UAV with a vertically polarized dipole working in the 3.1–4.8 GHz range was used for propagation measurements in three different scenarios. Initial measurements were performed between the UAV and receiver, assuming that no obstacles were present between the UAV and receiver. During this experiment, the receiver antenna at ground level was vertically polarized in one case and horizontally polarized in the other. By always leaving the transmitter located on the UAV vertically polarized, the VV and VH cases were produced. For the VV condition, a path loss of 72 dB and a path loss of 80 dB were observed for the VH condition. This indicates that there was an additional path loss resulting from the polarization mismatch. For ATG propagation measurements, EM waves are attenuated by multipath components, such as reflections, diffractions, and scattering. To estimate these losses, using the FSPL, two–ray [69], and log–distance path loss models may not be accurate in certain scenarios. Alternatively, empirical models, such as multi–slope log–distance path loss models [70], height–dependent two-ray models [71], and excess path loss models [72], are more reliable.

The models that have been addressed so far are all deterministic. These models do not consider the dielectric properties of obstacles that attenuate the signals. In such instances, statistical models such as log–normal shadowing are used to calculate the attenuation of signals due to random variations. There are two crucial variables in a log–normal shadowing expression, *μ_ψB_*: the mean of the random variable, and *σ_ψB_*: its standard deviation. A DJI N3 UAV, in combination with a *λ*/4 monopole, was used in [73] to measure path loss while the UAV moved from 0 to 24 m in height. In the context of path loss measurements, [73] proposed an altitude–dependent propagation loss model based on a zero-mean-behavior random variable. Under NLOS conditions, after performing propagation measurements at 1 GHz and 4 GHz, the *σ_ψB_* [74] value increased with the frequency and distance of the UAV from the receiver. The typical range of *σ_ψB_* is 5–12 dB for terrestrial macrocells and 4–13 dB for terrestrial microcells. For aerial wireless channels, *σ_ψB_* ranges from 1.2 to 5.24 dB, and it is observed that when UAVs fly at high altitudes, *σ_ψB_* can be as low as 1.2 dB [75].

### 3.2. UAV–Based Small–Scale Propagation Measurements

In [76], to investigate the dependence of small–scale fading on the altitude of a UAV, a hexacopter was equipped with a circularly polarized cloverleaf wire antenna, as shown in Figure 15. The receiving system consisted of a magnetic mount wideband high–frequency (MGRM–WHF) antenna, which is independent of the ground plane and was installed on a mast at the ground station, 1.5 m above the ground. The test environment consisted of multiple buildings and metal containers with the UAV taking a vertical path ranging from 0 to 100 m in height and a horizontal path maintaining 20–60 m from the receiver at ground level. The path loss exponents (PLEs) were estimated by varying the height of the UAVs to determine the relationship between small–scale fading and the UAV altitude. Furthermore, small–scale fading calculations are categorized into LOS and NLOS conditions. The Rician-K factor was utilized in the LOS case to explain the fading behavior. Adding the height parameter of a UAV to the Rician–K factor provided a better understanding of the small–scale fading. At lower altitudes, multipath components [77] from buildings and metallic containers combined vectorially at the receiver, causing fading. The cumulative distribution functions (CDF) [78] estimated small–scale fading in both the LOS and NLOS conditions.

In [75], a DJI N3 six–rotor UAV, equipped with a *λ*/4 monopole, was used to determine the fading depth, using UAVs by varying their vertical paths from 0 to 24 m in height, with a receiving station positioned 25 m away from the UAV. A *λ*/4 monopole with a ground plane and gain of 5.2 dBi, connected to a portable signal generator, enabled the continuous transmission of signals. We note from the measurements that the fading depth was independent of the operation frequency, which was more evident for the LOS conditions than for the NLOS conditions. A distribution function, such as the Nakagami, Rayleigh, Weibull, or Gaussian function, can describe the fading amplitude. By maintaining a root mean square error (RMSE) as low as 0.02 dB for both the LOS and NLOS scenarios, the log–logistic function [79] is the best distribution function among the available distribution functions.

In [80], to investigate the scattering effect of the buildings, ATG propagation measurements were conducted using a custom–built UAV equipped with a mmWave conical horn antenna (operating in the 26–40 GHz range) configured as the receiver and a planar elliptical dipole ultra–wideband (UWB) antenna (operating in the 3.1–5.3 GHz range) configured as the transmitter. In contrast, the ground station featured an mmWave conical horn antenna as the transmitter and a UWB antenna as the receiver. These ATG propagation measurements were instrumental in understanding the propagation characteristics of urban environments. This study presented power angle profiles for ATG propagation, which showed that in urban areas the presence of building rooftops causes a reduction in signal strength due to scattering. Additionally, the power elevation profile results indicated that when the UAV was at a higher altitude (50 m), the dominant propagation mechanism was due to reflections from buildings located behind the ground station. Measurements of outdoor–to–indoor coverage, conducted with the UAV hovering outside a building and the ground station positioned inside the building, revealed significant losses as the signals propagated through the building walls at both mmWave and UWB frequencies.

In ATG channels, which consist of a wireless channel between the UAV and the ground system, knife–edge diffraction (KED) is a commonly employed method for estimating the signal strength attenuation caused by diffraction. In KED, the EM wave diffracted by the building corners is determined by considering the obstacles to be thin and perfectly absorbing. The magnitude of the diffraction losses is calculated using mathematical formulas that consider Fresnel diffraction parameters. According to the UTD, diffraction losses are estimated using wedge geometry, which involves the wedge angle and reflection coefficient of obstacles and empirical models such as the linear regression model and the creeping wave linear model [81]. In [82], to understand the accuracy of KED and the empirical models, diffraction loss measurements were performed over a roof top in urban environments at 28 GHz. The measurement setup consisted of a transmitter antenna with a beamwidth of 10° and a receiver antenna with a beamwidth of 30°. The measurements were conducted at two sites to understand the influence of the TX distance from the LOS boundary and the RX distance from the LOS/NLOS boundary. The study found that the diffraction losses increased when the distance from the diffraction edge increased and decreased when the distance between the TX and the building decreased. The loss was shown to be proportional to the diffraction angle.

Using UAV–based propagation measurements, we can estimate the attenuation of the signal when the transmitter follows vertical and horizontal paths. In contrast, conventional methods fail to evaluate diffraction losses and multipath components from the corners and edges of buildings, which are typically between 10 m and 25 m in height. In [83], horn antennas with a gain of 20 dBi and an HPBW of 17° were used indoors and outdoors for propagation measurements. The indoor measurements were analyzed using three types of wall construction: plastic boards, wooden walls, and dry walls. During the measurements, the receiving and transmitting horns were placed at a 1.4 m height above the corner of the wall. The measurements were performed at 10 GHz, 20 GHz, and 26 GHz. The measured data were compared with the theoretical estimates using the KED model. The practical measurements were in good agreement with the theoretical calculations for a dry wall. However, the KED overestimated losses by 2–4 dB in the case of wooden walls and plastic boards. For outdoor measurements, it was found that KED accurately calculated the diffraction losses for sharp edges, whereas linear models using a minimum mean square error (MMSE) linear fit derived from actual measured data were more accurate for rounded edges.

To investigate the scattering effects of buildings, we created an ATG propagation scenario. In this setup, two buildings with heights of 10 m and 20 m were modeled using CST Studio Suite. Building 1 was modeled with a height of 20 m while Building 2 was modeled with a height of 10 m. A horn antenna configured as a receiver was mounted on a mast placed on top of Building 2 to identify potential scattering regions. In the simulation environment, we modeled a UAV equipped with a horn antenna flying at a height of 17 m above ground level. The complete simulation setup, including the scattered rays, is shown in Figure 16. The two buildings were positioned 30 m apart in this scenario. The structure was analyzed using an asymptotic solver based on the shooting and bouncing ray (SBR) technique, which allowed us to observe how signals were diffracted at the corners of the buildings. The SBR technique provides an initial estimation of ATG propagation. However, to accurately understand scattering effects in real–world scenarios, practical UAV–based propagation measurements are necessary. In such measurements, a UAV equipped with a transmitter antenna and a portable signal generator would be used. On the ground, a receiver setup consisting of a horn antenna mounted on a mast and a spectrum analyzer connected to the antenna would be used to calculate the power levels of the received signal.

Several propagation measurements were carried out in [84] using a hexacopter equipped with a narrowband antenna resonating at 440 MHz and a wideband antenna operating between 1 GHz and 6 GHz. The measurements were performed in a suburban area at 440 MHz and 1 GHz. The UAV flew in a vertical path with an altitude of 0–25 m over two buildings of 15 m and 25 m in height. Although there were other obstacles, such as trees and cars, in addition to the two main buildings, the diffraction losses owing to other obstacles were minimal at high altitudes. A comparison was made between UAV–based propagation measurements and theoretical modeling, such as the KED model. In general, the measurements by UAVs and the theoretical calculations are in good agreement at lower frequencies; however, at higher frequencies, the diffraction losses are more significant, and the theoretical calculations underestimate these losses.

### 3.3. Selecting the UAV Antenna

Choosing an appropriate antenna for UAV–based measurements is essential before conducting measurements. To ensure that the UAV–based in situ measurements are accurate, it is important to calibrate the antenna before mounting it on the UAV. Several factors are considered when selecting a UAV antenna: it should be compact, lightweight, mechanically stable, unaffected by wind, and electromagnetically insensitive to the structure of the UAV. Because of the several metallic components on the body of the UAV, directional antennas are likely to experience EM coupling with the UAV body, which can degrade its performance.

To understand the EM behavior of the antenna [85] mounted on the UAV, we simulated a complete UAV structure using the CST Studio Suite. Figure 17 depicts a DJI F450 UAV equipped with a pyramidal horn antenna simulated at 8 GHz using the SBR technique. Apart from the main beam from the horn, there is a portion of signals scattered from the UAV body, which can create errors in antenna measurements. This explains the necessity for care to be taken before selecting an antenna and understanding its behavior after mounting [86] it on the UAV [87]. There are several ways to mitigate the effects of scattering, including changing the antenna design [87,88], optimizing the antenna location [89], and using RF absorbers in areas where the UAV body exhibits potential reflections. Whenever we choose antennas for UAVs, there is always a tradeoff between an antenna with a narrow beamwidth and an antenna with wider beamwidth. Antennas with a wider beamwidth cover wide angles, which means that small deviations in the alignment of the UAV relative to the AUT have a less pronounced effect. The signal remains closer to the intended polarization, minimizing the introduction of unwanted cross–polarized components. However, with an antenna with a wider beamwidth, there will be high scattering from UAVs, affecting the co–polarization and cross–polarization patterns.

On the other hand, antennas with narrow beam widths have an advantage in terms of low scattering from the UAV body. However, they have some limitations as well. The major challenge for these antennas is alignment between the antenna mounted on the UAV and the AUT. In the case of an antenna with a narrow beamwidth, vibrations from the UAV body can create a misalignment between the antenna mounted on the UAV and the boresight of the AUT. To minimize misalignment errors, additional efforts must be made to maintain the antenna’s beam peak at the AUT’s boresight throughout its trajectory.

Table 3 presents the different antennas used in the literature. An omnidirectional or directional antenna was used depending on the area of application. These antennas are specially designed for UAV applications, considering beamwidth and radiation pattern constraints. In cases such as dipole [90] and helix [91] antennas, the ground plane is included as part of the antenna. The commonly used antennas with directional or omnidirectional patterns have limitations. Directional antennas are prone to misalignment errors; hence, additional precautionary steps are required to overcome them. Omnidirectional antennas have the limitations of low gain. On the other hand, in [92], an array of half-bowtie antennas was designed to cover all hemispherical regions. This design has better coverage with an HPBW of 240° in the azimuth and 98.6° in elevation, similar to an omnidirectional antenna; with a gain of around 5.9 dBi.

### 3.4. Accuracy Analysis of UAV–Based Antenna Measurement

The accuracy of the UAV-based antenna measurements mainly depends on the accuracy of the RF equipment mounted on the UAV, the amount of vibration experienced by the UAV, the accuracy of the GPS positioning, and the external environmental conditions. To understand how these aspects affect the measurements, Table 4 describes the variations in the radiation patterns measured by the UAV. All these measurements were performed by a UAV to characterize an antenna installed outdoors. After characterizing the antenna in terms of the radiation pattern, the same antenna was measured in anechoic chambers, and in some cases, it was simulated using commercially available EM solvers. As described in Table 4, it can be understood that due to the UAV vibrations, external environmental conditions, and drifting in the UAV positions, a maximum error of 1 dB in the peak amplitude is noticed. These results indicate that UAV measurements are an accurate and a reliable solution for characterizing an antenna.

In UAV–based antenna measurements, the radiation patterns of the antenna in either the azimuth or elevation plane were obtained by following a predefined trajectory. For the UAV to follow this predefined trajectory, the UAV path planning is achieved through software tools such as QGroundControl v1.3.8 [103]. The accuracy with which the GPS follows the defined waypoints depends on the accuracy of the GPS used on the UAV. Based on the frequency of operation, GPS systems such as differential RTK (D–RTK), RTK, differential GNSS, and real–time differential GPS are used. These high–precision systems are particularly used for UAV measurement applications, which can provide centimeter and sub–meter accuracy. Based on the UAV–based antenna measurements conducted in [27], it is evident that a total deviation of 0.38 dB between the UAV–based measurements and the measurements in the anechoic chamber is observed. Out of the 0.38 dB variation in the peak amplitude of the radiation patterns, 0.36 dB is due to the external environmental conditions; small inaccuracies in D-RTK positioning result in a 0.01 dB variation, and variations in RF component behavior due to outdoor temperature result in a 0.02 dB variation. In [7], the horizontal deviation between the trajectory followed by the UAV and the trajectory planned in the software was less than 2 m. These deviations were due to environmental conditions such as wind; in any case, these effects resulted in an angular deviation of less than 0.38° in the UAV measurements. In [101], small deviations in the trajectory followed by the UAV resulted in a deviation of 0.02 dB, and the variations in the relative orientation of the UAV could produce an uncertainty of ±2%, producing a variation of 0.005 dB in the UAV measured results. Similarly, in [50], differential GNSS, which can provide an accuracy of only sub–meter level accuracy, was used. Although a centimeter–level accuracy GPS such as D-RTK was not used here, a deviation of 0.03° was observed in the UAV measurements due to the deviations in the UAV positioning. This is because the measurements were performed at 175 MHz; in this case, such low frequencies do not demand centimeter–level accuracy in UAV positioning. In [99], where UAV measurements were performed at 8–12 GHz, D–RTK was used for UAV positioning, and deviations in the UAV positioning resulted in a variation of ±0.01° in the UAV measurements.

From these analyses of the accuracy of UAV–based antenna measurements, it is evident that factors such as GPS positioning, vibrations in UAV, and changes in UAV alignment due to external environmental conditions result in relatively small deviations in the radiation pattern of the antenna. However, in [42,102], major deviations in UAV measurements and measurements from anechoic chambers were observed. This is due to the ground reflections that account for deviations in the antenna parameters, such as side-lobe level and HPBW, and, in certain scenarios, in the peak gains. Ground reflections create multipath interference, where signals add constructively or destructively before reaching the receiver antenna [104]. Based on the constructive or destructive interference, ground reflections lead to either an increase or decrease in the measured antenna gain.

To mitigate the effects of ground reflections, strategies such as those using radiation-absorbent material or diffraction fencing can be employed. Radiation–absorbent material, simply known as an absorber, is used in anechoic chambers, which helps to reduce interferences due to reflection from the ground. However, several absorbers would be required for outdoor measurements to mitigate the ground reflections; this is impractical and costly. Another approach is to use a metallic diffraction fence to block the ground–reflected waves. This approach is applicable and used in elevated slant measurement ranges [105].

To overcome ground reflections, we propose the use of two UAVs, which allows the AUT and receiving antenna mounted on the UAV to maintain higher altitudes from the ground level. In the two proposed UAV antenna measurements, one of the UAVs is configured as a transmitter consisting of a portable signal generator and an AUT. Similarly, the second UAV, which is configured as a receiver, consists of a real–time spectrum analyzer and an antenna to receive signals from the transmitter UAV. The proposed solution enables non–tethered UAV operation, allowing UAVs to maintain a higher altitude from the ground level, thus avoiding the effects of ground reflections. By adopting two UAV antenna measurements and placing the antennas at appropriate locations on the UAV to avoid electromagnetic coupling, antenna measurements with good accuracy can be performed in outdoor environments.

## 4. Conclusions

In this article, we presented a comprehensive review of UAV–based antenna and propagation measurements, offering a detailed analysis of the various factors influencing these measurements. The study provides a set of guidelines for selecting the UAV antennas, ensuring higher levels of measurement accuracy. Additionally, we compared traditional slant–range methods with innovative, low–cost UAV test setups and explored the extension of path loss models by incorporating UAV altitude as a critical parameter.

The discussion included several practical test cases, such as the use of parabolic reflector systems on ships, BASTAs, LFAAs, one of the world’s largest radio telescopes, and ONERA’s Nostradamus system, which features 288 biconical antennas operating in the HF range (3–30 MHz). We also examined propagation measurements for both large–scale and small–scale channel effects.

The findings suggest that advancements in portable devices like SDRs, high–precision positioning systems with centimeter–level accuracy, custom antenna designs, and UAVs constructed from lightweight and durable materials such as carbon fiber have significantly expanded the potential for UAV–based antenna and propagation measurements. For applications in 5G/6G, where accuracy is paramount, UAV–based test setups have emerged as the preferred measurement solution.

In conclusion, we anticipate that this review will serve as a valuable reference for the further development of UAV–based measurement solutions, driving innovation and precision in the field.

## Figures and Tables

**Figure 1 sensors-24-07395-f001:**
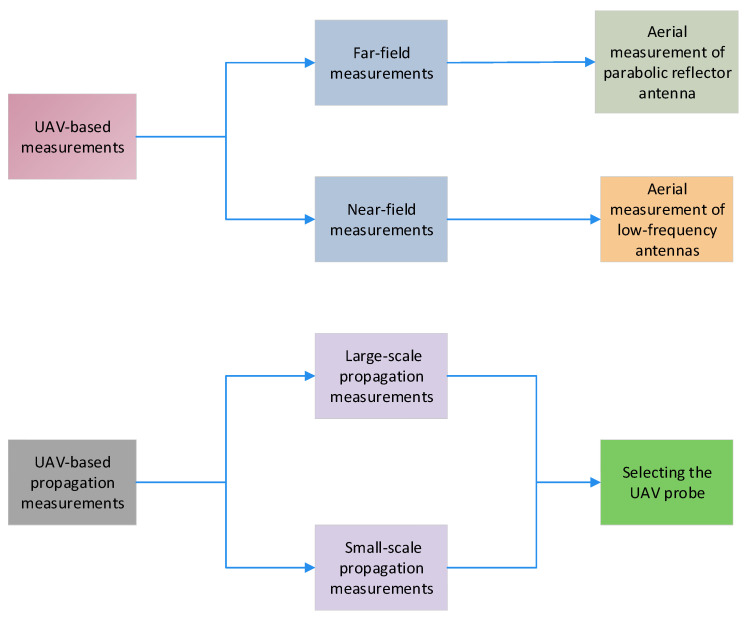
Organization of the paper.

**Figure 2 sensors-24-07395-f002:**
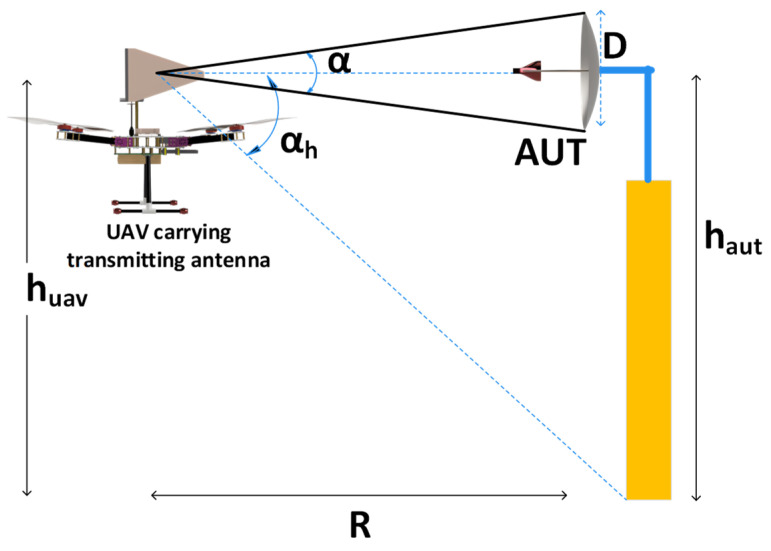
Measurement configuration of the UAV system.

**Figure 3 sensors-24-07395-f003:**
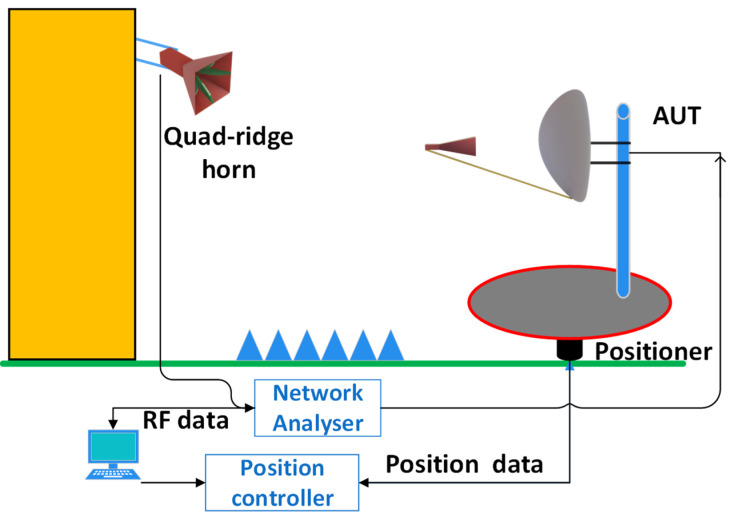
Conventional elevated slant test range.

**Figure 4 sensors-24-07395-f004:**
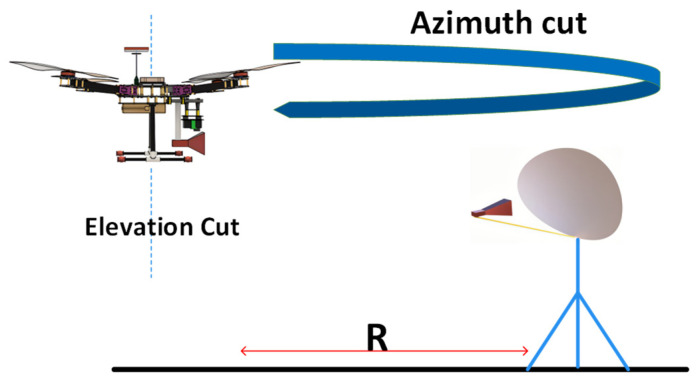
UAV–based in situ measurement for a parabolic reflector antenna system.

**Figure 5 sensors-24-07395-f005:**
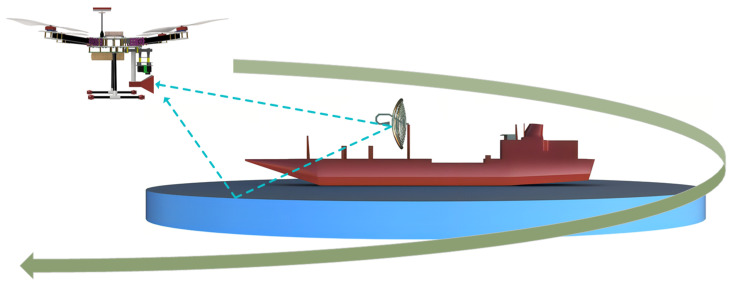
UAV–based measurement for a parabolic reflector antenna system placed on a ship.

**Figure 6 sensors-24-07395-f006:**
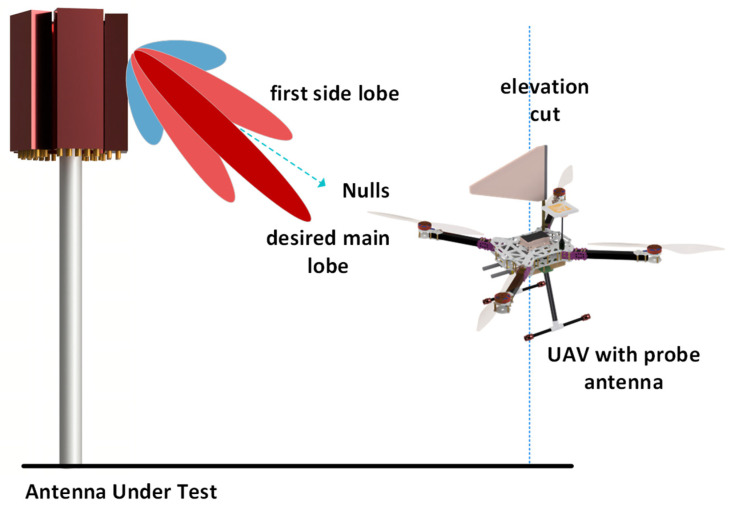
Vertical radiation pattern of a BASTA using a UAV.

**Figure 7 sensors-24-07395-f007:**
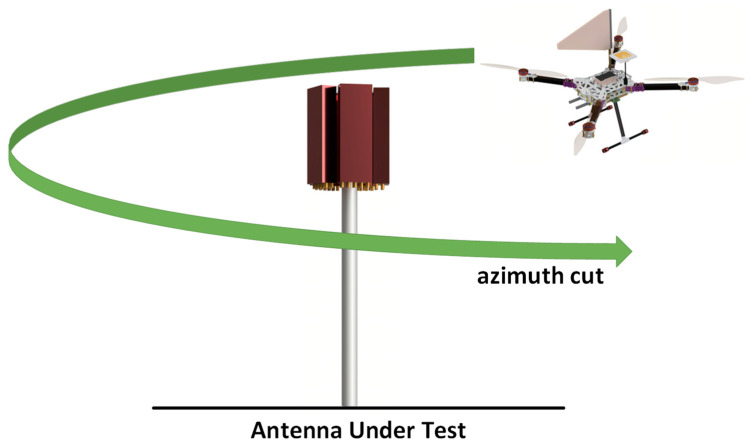
Horizontal radiation pattern of a BASTA using a UAV.

**Figure 8 sensors-24-07395-f008:**
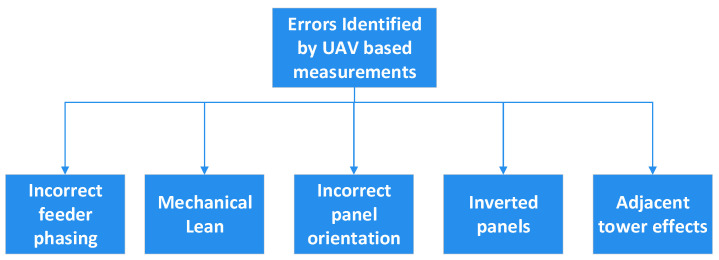
Common errors in broadcasting systems.

**Figure 9 sensors-24-07395-f009:**
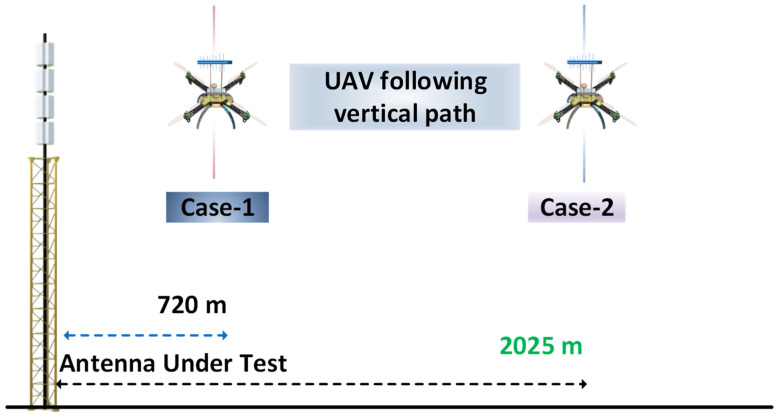
SixArms airborne measurements for broadcasting systems.

**Figure 10 sensors-24-07395-f010:**
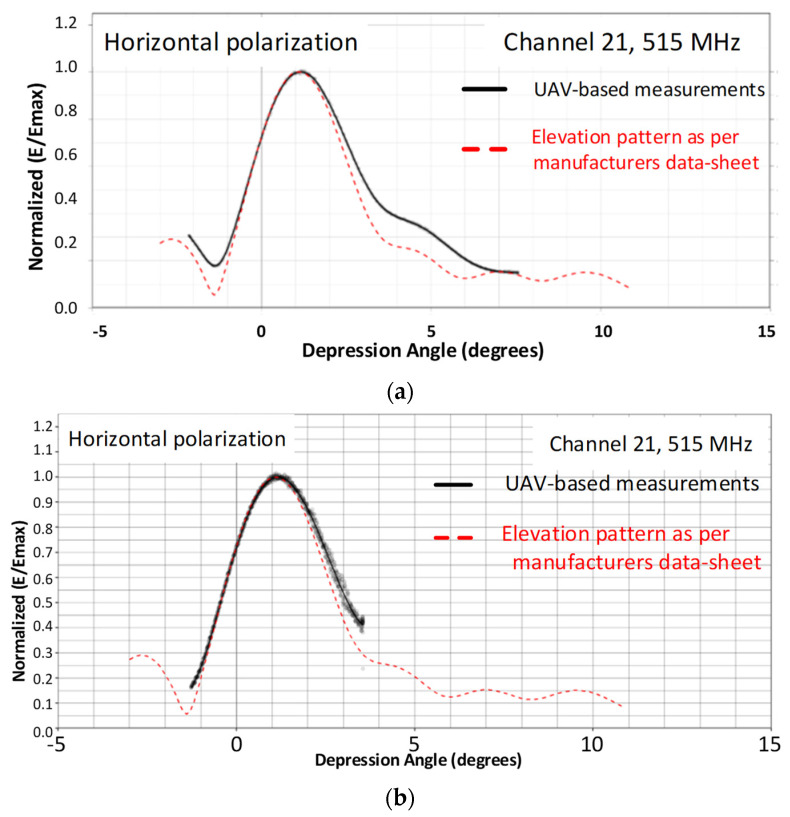
(**a**) UAV–based measurements at 720 m and (**b**) UAV–based measurements at 2025 m.

**Figure 11 sensors-24-07395-f011:**
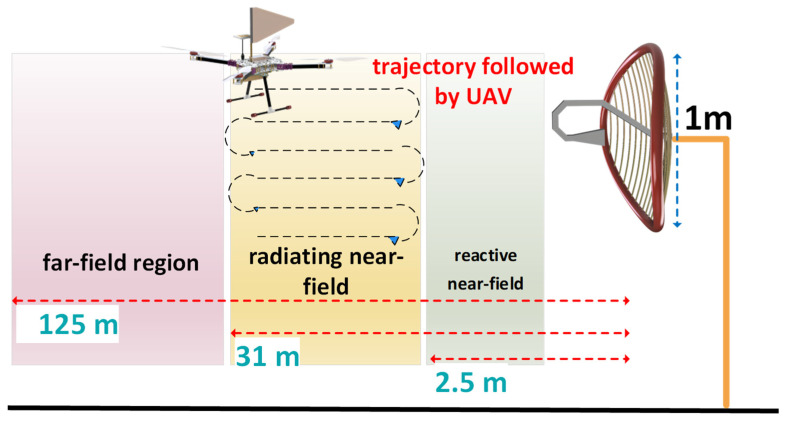
UAV–based measurements in radiating near field.

**Figure 12 sensors-24-07395-f012:**
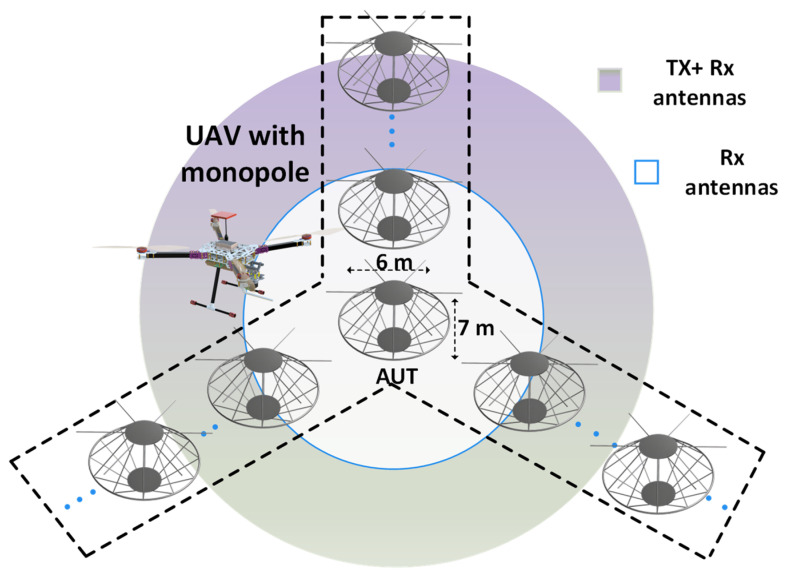
UAV–based measurements for an array of HF wire biconical antennas.

**Figure 13 sensors-24-07395-f013:**
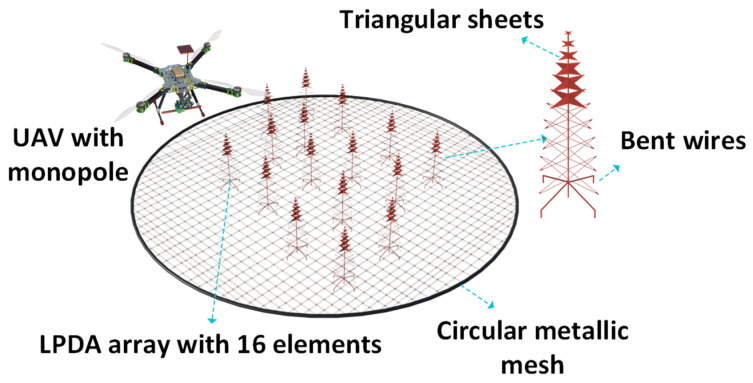
UAV with monopole flying over the LPDA array.

**Figure 14 sensors-24-07395-f014:**
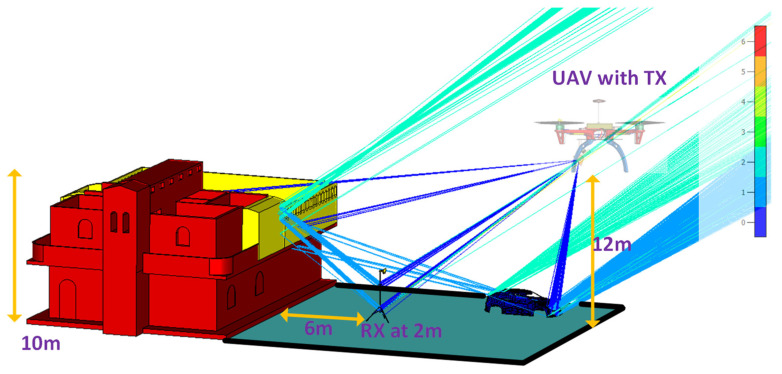
Scattering effect in a semi–urban area simulated in CST.

**Figure 15 sensors-24-07395-f015:**
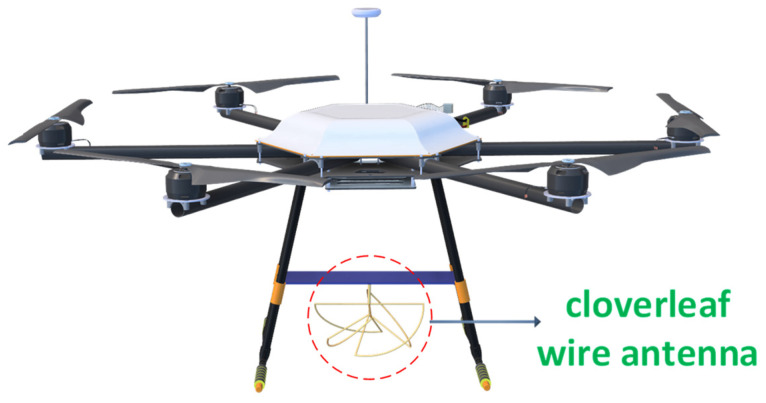
Hexacopter carrying cloverleaf wire antenna.

**Figure 16 sensors-24-07395-f016:**
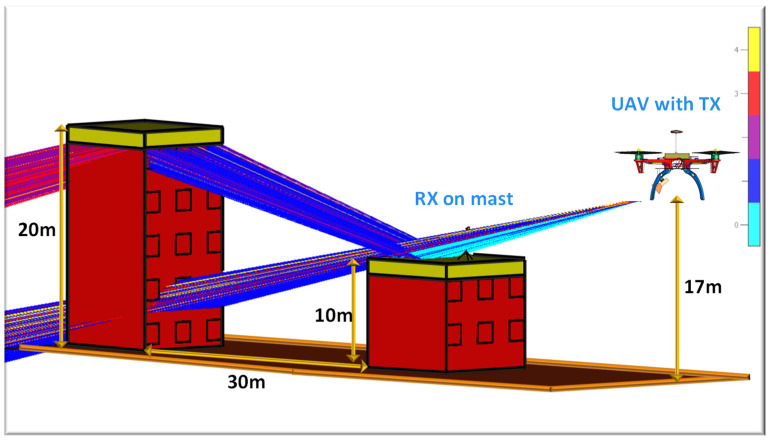
Scattering effect in an urban area simulated in CST Studio Suite.

**Figure 17 sensors-24-07395-f017:**
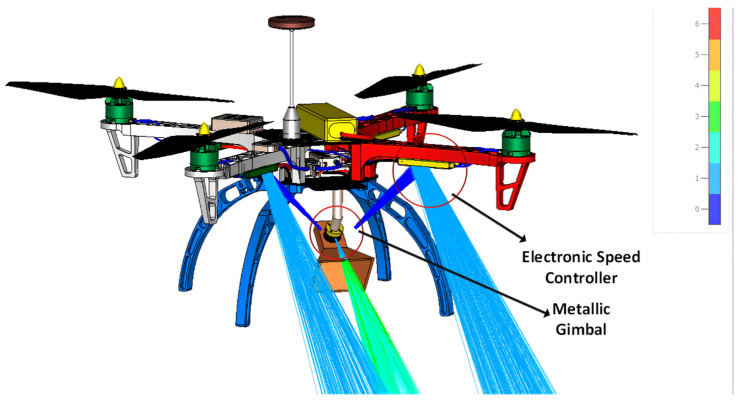
Fields scattered by the UAV body.

**Table 1 sensors-24-07395-t001:** Far–field measurements.

Reference	Frequency of Operation	Far–Field Distance	AUT
[27]	14.5 GHz	350 m	Parabolic reflector.
[28]	9.5 GHz	60 m	Reflector mounted on ship.
[35]	515 MHz	2025 m	Broadcasting antenna.

**Table 2 sensors-24-07395-t002:** Near–field measurements.

Reference	Frequency of Operation	Near–Field Distance	AUT
[41]	4.65 GHz	3.4 m	Offset reflector
[52]	20 MHz	4 m	Ground plane antenna
[50,51]	175 MHz	15 m	SKALA

**Table 3 sensors-24-07395-t003:** State–of–the–art antennas for UAV applications.

Reference	Antenna	Frequency Range (GHz)	Radiation Pattern
[21]	Micro–Strip Patch	1.8–2.7	Directional
[87]	Micro–Strip Patch	2.4–5.2	Directional
[93]	Log–periodic meandered dipole array	0.85–2.2	Directional
[94]	Vivaldi	1.5–4.5	Directional
[95]	Horn	7.5–18	Directional
[90]	Dipole	0.55–1.6	Omnidirectional
[96]	Surface Wave	6.1–18	Directional
[91]	Helix	0.6–1.1	Omnidirectional
[97]	Quasi–Yagi	23–28.5	Directional
[92]	Half–bowtie antenna	4.1–5.6	Directional

**Table 4 sensors-24-07395-t004:** Accuracy of UAV–based antenna measurements.

Reference	Frequency (GHz)	Difference Between UAV and Anechoic Chamber Measurements	Difference Between UAV and Simulation Results
[1]	0.75	NA	0.5 dB in peak amplitude
[4]	0.47 to 0.7	NA	0.6 to 1 dB in peak amplitude
[7]	0.7 to 0.8	NA	0.5 to 1 dB in peak amplitude
[27]	14.5	0.38 dB in peak amplitude	NA
[42]	4.65	Widening in radiation pattens	NA
[50]	0.175	NA	<1 dB in peak amplitude
[98]	0.05 to 0.35	NA	0.5 to 1 dB in peak amplitude
[99]	8 to 12	0.5 dB in peak amplitude, 0.06° in HPBW	NA
[100]	0.05 to 0.32	NA	<0.1 dB in peak amplitude and 1 dB in SLL
[101]	44	1 dB in peak amplitude	NA
[102]	4 to 6	3 dB in peak amplitude	NA

## Data Availability

Not applicable.

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
