# Peer review of "A Review of Unmanned Aerial Vehicle Based Antenna and Propagation Measurements"

_sensors, 2024, doi:10.3390/s24227395_

Round 1

Reviewer 1 Report

Comments and Suggestions for Authors

The article is written to summarize the current state of antenna and propagation measurement techniques based on UAVs. The article has interpreted and presented relevant results correctly. But, in my opinion, the measurement techniques presented in this paper just describe how the UAVs with TR/RX antenna can be used to replace those in traditional setups of measurement systems. There is barely anything new in the measurement theory or mechanism. In addition, it seems the measurement accuracy may be limited by the UAVs, such as the shaking or unsmooth moving of the UAVs. A deep analysis is desired instead of a straight state.

Author Response

In addition, it seems the measurement accuracy may be limited by the UAVs, such as the shaking or unsmooth moving of the UAVs. A deep analysis is desired instead of a straight state.

Authors Response: We thank the reviewer for bringing this point to our attention. The reviewer is correct that when performing measurements with UAVs, it is possible that UAV vibrations can alter the UAV position, affecting the accuracy of the measurements. To understand how aspects such as UAV vibrations, drift in UAV position due to external wind, and errors in GPS positioning can affect the radiation patterns measured by UAV, we have presented an in-depth analysis. These aspects are described in Table 4, in the revised manuscript. Moreover, relevant explanations are provided in Section 3.4. 

Reviewer 2 Report

Comments and Suggestions for Authors

The authors in this paper have presented a comprehensive survey of the state-of-the-art in UAV-based antenna and propagation measurements. The topic is interesting and the discussions are well organized. Here, we have summarized some comments as follows:

1. It is fine that the authors have cited some impacting papers related to the UAV-based antenna and propagation measurements. However, some parts need to be improved. For example, in introduction, some newly published related to the UAV communications need to be cited and discussed, such as “A UAV-aided real-time channel sounder for highly dynamic nonstationary A2G scenarios”, “Physics-based 3D end-to-end modeling for double-RISs assisted non-stationary UAV-to-ground communication channels”, and “A survey on channel sounding technologies and measurements for UAV-assisted communications”, et al.

2. The figures of this paper need to be significantly improved. For the provided figures, it is difficult for the reviewer to see the symbols very clearly.

3. The language written of this paper needs to be carefully improved, in fact, we can easily notice many grammar errors here and there this paper.

Comments on the Quality of English Language

The language written of this paper needs to be carefully improved, in fact, we can easily notice many grammar errors here and there this paper.

Author Response

It is fine that the authors have cited some impacting papers related to the UAV-based antenna and propagation measurements. However, some parts need to be improved. For example, in introduction, some newly published related to the UAV communications need to be cited and discussed, such as “A UAV-aided real-time channel sounder for highly dynamic nonstationary A2G scenarios”, “Physics-based 3D end-to-end modeling for double-RISs assisted non-stationary UAV-to-ground communication channels”, and “A survey on channel sounding technologies and measurements for UAV-assisted communications”, et al.

Authors Response: We thank the reviewer for bringing this to our attention. The reviewer is correct that recent papers on UAV measurements can be cited in the Introduction. In the revised manuscript, we have cited “A UAV-aided real-time channel sounder for highly dynamic nonstationary A2G scenarios” and “Physics-based 3D end-to-end modelling for double-RIS assisted non-stationary UAV-to-ground communication channels,” as suggested by the reviewer. 

The figures of this paper need to be significantly improved. For the provided figures, it is difficult for the reviewer to see the symbols very clearly.

Authors Response: We appreciate the reviewer for highlighting this matter. To make the figures readable, we updated the font sizes of the symbols and legends in Figs. 2, 4, 10, 11, 12, 13, and 15 of the revised manuscript.

The language written of this paper needs to be carefully improved, in fact, we can easily notice many grammar errors here and there this paper.

Authors Response: We have thoroughly proofread the whole manuscript to correct any grammatical errors.

Reviewer 3 Report

Comments and Suggestions for Authors

This paper handles propagation measurements with Unmanned Aerial Vehicles (UAVs) as powerful tools for in-situ electromagnetic field assessments.

I have major remarks on the presented paper.

- It is unclear to me what the own scientific contributions are. The paper lacks any novelty and own work.

- What audience you are focussing at? RF engineers? UAV drone pilots? For both target groups this paper only lists work of others.

- An abstract is not a summary of the work. Also conclusions should be added in the abstract.

- Are Figures 3, 9, 15, 16 relevant? As they only visualize a regular list of different items?

Author Response

It is unclear to me what the own scientific contributions are. The paper lacks any novelty and own work.

Authors Response: We would like to take this opportunity to clarify the novel aspects of our work and highlight the original contributions. In this paper, we have provided state-of-the-art UAV-based antenna measurements. However, in the earlier version, we did not highlight any research gaps. In the revised manuscript, we have performed an error analysis on UAV-based antenna measurements, while future research directions are proposed. Moreover, based on the state-of-the-art study, it was identified that configuring two UAVs—one as a transmitter and the other as a receiver—provides greater flexibility in measurements. With the proposed methodology, configuring two UAVs enables them to reach higher altitudes above ground level. This allows UAV-based antenna measurements to be performed while minimizing the effects of ground reflections.

What audience you are focussing at? RF engineers? UAV drone pilots? For both target groups this paper only lists work of others.

Authors Response: This paper is focused on UAV-based measurements, specifically targeting RF engineers. In fact, it is the first review paper to cover UAV-based measurements, encompassing both near-field and far-field techniques.

An abstract is not a summary of the work. Also conclusions should be added in the abstract.

Authors Response: The abstract has been updated in the revised manuscript.

Are Figures 3, 9, 15, 16 relevant? As they only visualize a regular list of different items?

Authors Response:  We thank the reviewer for his/her valuable suggestions. In response, we have removed Figs. 3, 15, and 16. In the revised manuscript, we have replaced the block diagrams with relevant descriptions.

Round 2

Reviewer 3 Report

Comments and Suggestions for Authors

Dropping new items in the abstract or adding a new paragraph does not mean that my earlier remarks are solved. Scientific conclusions are still missing in the abstract. In the section "Accuracy analysis of UAV‐based antenna measurement" new concepts are given, without really explaining them and without linking them to the rest of the scientific work. Also the focus of audience is still unclear for me.

Author Response

Dropping new items in the abstract or adding a new paragraph does not mean that my earlier remarks are solved. Scientific conclusions are still missing in the abstract.

 In the section "Accuracy analysis of UAV‐based antenna measurement" new concepts are given, without really explaining them and without linking them to the rest of the scientific work.

Also the focus of audience is still unclear for me.

Authors Response: We thank the reviewer for their valuable suggestions. This paper is a Survey paper focused on researchers performing antenna measurements both in industry and in academia. We aim to present and compare available solutions to measure antennas in outdoor conditions. This includes measuring structurally large antennas, which are relatively difficult to fit in an indoor anechoic chamber, and performance evaluation of wideband antennas in outdoor conditions. In response to the reviewer’s suggestion, we have modified the abstract in the updated manuscript. In the updated abstract, we have highlighted the scientific conclusions, such as the techniques to mitigate ground reflections, to calculate building diffraction loss, and to simulate the effect of the UAV body on the antenna measured. For the new section included “Accuracy analysis of UAV-based antenna measurements” in the updated manuscript, we have provided descriptions about software tools used for trajectory planning. Also, we have explained in more detail the need to mitigate ground reflections with various possible solutions. Finally, we have concluded the paper by highlighting two UAV measurement solutions that help avoid ground reflections.